# A multi-functional nine channels full-spectrum light emitting diode color temperature palette

Jingjing Liang[1,2], Hui Cao[1]*, Donghua Xu[3], Xiaobing Lu[4,5], Feijiang Huang[1], Javid Atai[6]

**1** School of Computer Science (School of Artificial Intelligence), Guangzhou Maritime University, Guangzhou, China, **2** School of Physics and Materials Science, Guangzhou University, Guangzhou, China, **3** Public laboratory center, Guangzhou Maritime University, Guangzhou, China, **4** Department of Nutritional and Metabolic Psychiatry, The Affiliated Brain Hospital of Guangzhou Medical University, Guangzhou, China, **5** Guangdong Engineering Technology Research Center for Translational Medicine of Mental Disorders, Guangzhou, China, **6** School of Electrical and Computer Engineering, The University of Sydney, New South Wales, Australia

* 978739471@qq.com

## Abstract

A multi-functional nine-channel full-spectrum light emitting diode color temperature palette is proposed. The color temperature (CCT) of the palette can be continuously tuned at 1K interval within the range of 2500K—6500K. A high color rendering index (CRI) is achieved, with Ra > 94 and R1—R15 > 90. A fidelity index (Rf) greater than 91 is obtained, and the gamut index (Rg) within the range of 96 ≤ Rg ≤ 104 is achieved. Additionally, A chromaticity deviation (Duv) within the range of |Duv| < 0.001 is realized. The circadian effect with melanopic efficacy of luminous radiation(MELR) of 0.6 mW/lm—1.14 mW/lm and circadian action factor(CAF) of 0.289 blm/lm—0.656 blm/lm are obtained. To verify the theoretical model's stability, the Gaussian distribution function of monochromatic LED spectrum is adjusted to induce a 5% asymmetry in integral areas, and the FWHM is offset by ±10%, simulating the non-ideal conditions in practical applications to assess the model's robustness. The color temperature palette enables precise control and reproduction of CCT of sunlight with corresponding circadian effect, which is of positive significance for visual fields such as photography or art exhibitions and regulation of circadian rhythm.

## Introduction

In recent years, full-spectrum lighting has been steadily gaining traction in the lighting market. Renowned for its exceptional color rendering capabilities, it is the preferred choice for applications demanding high-fidelity color representation, including cinematography, art galleries, and exhibitions [1–3]. The discovery of intrinsically photosensitive retinal ganglion cells (ipRGCs) [4–7] has propelled healthy lighting to the forefront of research. Full-spectrum light sources distinguish themselves by their capacity to replicate the natural light environment and adjust CCT according to the time of day. In the solid-state lighting era, LEDs offer distinct advantages, such as high

**Data availability statement:** All relevant data are within the paper and its Supporting Information files. The code and data used in the article can be accessed through this link: https://doi.org/10.5281/zenodo.15517037.

**Funding:** This study was supported by the following funders: Bureau of Education of Guangzhou Municipality (202234641, received by Dr. Hui Cao); Department of Education of Guangdong Province (2020ZDZX3072, 2021KTSCX098; received by Dr. Hui Cao and Dr. Feijiang Huang, respectively); Education and Science Program of Guangdong Province (2022GXJK308, received by Dr. Feijiang Huang); the Project of Construction Discipline Scientific Research Capability Improvement of Guangdong Province (2022ZDJS098, 2022ZDJS096, 2024ZDJS055; received by Dr. Hui Cao, Dr. Feijiang Huang and Dr. Dan Xiang, respectively); the Research Project of Construction National Science and Technology Think Tank of Guangdong Province (SXK20220201035, received by Dr. Feijiang Huang); Guangzhou Key R&D Program Agriculture and Social Development Science and Technology Project (202206010034, received by Dr. Xiaobing Lu); Guangzhou Municipal Basic Research Project jointly funded by municipal schools (institutes) (2023A03J0834, received by Dr. Xiaobing Lu). All funders played a role in the research design, data collection and analysis, publication decisions, or manuscript preparation process. Specifically: - The Bureau of Education of Guangzhou Municipality (202234641), Department of Education of Guangdong Province (2020ZDZX3072, 2021KTSCX098), and Education and Science Program of Guangdong Province (2022GXJK308) contributed to research design, data collection, and analysis. - The Project of Construction Discipline Scientific Research Capability Improvement of Guangdong Province (2022ZDJS098, 2022ZDJS096, 2024ZDJS055) played a role in manuscript preparation. - The Research Project of Construction National Science and Technology Think Tank of Guangdong Province (SXK20220201035), Guangzhou Key R&D Program Agriculture and Social Development Science and Technology Project (202206010034), and Guangzhou Municipal Basic Research Project jointly funded by municipal schools (institutes)

energy efficiency and extended lifespan. Integrating with the concept of health lighting, full-spectrum LED lighting has emerged as a significant development [8]. It is worth mentioning that the current exploration of metal halide perovskite materials [9] and the attempt to obtain white light by combining the five different emission lights of organolead halide perovskites [10] have also added rich research perspectives to this field.

Currently, full-spectrum LEDs are primarily developed through two approaches: LED chips combined with phosphors [11] and multi-channel LED chips. In the early development of the phosphor+LED chip method, white LEDs were mainly produced using either a blue LED chip (440–470 nm) coated with $Y_3Al_5O_{12}:Ce^{3+}$ yellow phosphor, or a near-ultraviolet (n-UV) chip paired with tricolor phosphors (red+green+blue), as well as combinations of blue chips with green and red phosphors [12–16]. However, these methods suffered from the "cyan gap" issue [17], prompting researchers to prioritize the development of new phosphors emitting in the blue-cyan region [18–20]. Despite these efforts, this approach has inherent limitations, including low quantum conversion efficiency of phosphors, stringent manufacturing requirements for cyan phosphors, and challenges related to thermal quenching, water resistance, and thermal stability.

For the approach utilizing multi-channel LED chips, two key advantages are evident: high luminous efficiency and flexible CCT regulation. As these multi-channel white LEDs are directly driven by individual chip currents, energy losses from stokes shift and non-radiative recombination are eliminated, significantly reducing energy dissipation and enhancing luminous efficiency. Moreover, chromaticity, CCT, and other optical properties can be precisely controlled by adjusting the current of each LED chip, enabling CCT tuning within a specific range [21–25]. However, existing research indicates that CCT tuning typically features limited and discontinuous switching points. Light sources capable of smooth, 1K-interval CCT adjustment remain scarce, especially those that can simultaneously maintain high color rendering and non-visual effects.

To achieve high-quality illumination, light sources must meet stringent criteria. A minimum CRI of over 90 is essential for excellent color rendering, along with a small Duv. Additionally, short-wave blue light, known to damage the retina and disrupt circadian rhythms [26–27], must be carefully controlled in spectral design. With the growing understanding of light's non-visual effects, full-spectrum light sources that replicate daily sunlight CCT variations and support circadian regulation represent a highly promising research direction.

In this paper, a multi-functional nine-channel full-spectrum LED color temperature palette is proposed. Unlike common four-channel [21], five-channel [25], and six-channel [22] LEDs, nine-channel full-spectrum LED color temperature palette allows for continuous CCT tuning at 1K interval within the range of 2500K—6500K, enabling precise control and reproduction of sunlight's CCT variations. It is precisely this characteristic that makes it possible for this nine-channel LED to be regarded as a color temperature palette. By systematically adjusting the luminous flux ratios of the nine LEDs, high values for the CRI, Rf, Rg [28], and a small Duv value are achieved.

Beyond replicating sunlight's CCT changes, this color temperature palette is designed to mimic sunlight's non-visual effects, thereby regulating human circadian

(2023A03J0834) all played a role in the research design.

**Competing interests:** The authors have declared that no competing interests exist.

rhythms more effectively. To this end, the non-visual effect parameters MELR [24] and CAF [22]are optimized. The optimization process considers the maximum and minimum MELR values under Rf constraints [29] and the characteristics of light at different time of day.

## Materials and methods

Multi-channel LEDs operate based on the principle of spectral linear superposition. The overall spectral power distributions (SPDs) are composed of the SPDs of individual LED chips, and the mathematical representation is detailed below:

$$S(\lambda) = L_1 \cdot S_1(\lambda) + L_2 \cdot S_2(\lambda) + L_3 \cdot S_3(\lambda) + ... + L_n \cdot S_n(\lambda) \tag{1}$$

where $\lambda \in [380,780]$, $n \in [1,9]$; $S(\lambda)$, $L_n$ and $S_n(\lambda)$ are the overall SPDs, the luminous flux ratio and the SPD of the n-th monochromatic LED, respectively. We assume that the SPD of the n-th monochromatic LED follows the Gaussian distribution [30], which can be expressed by:

$$S_n(\lambda) = \frac{1}{\sigma\sqrt{2\pi}} e^{-\frac{1}{2}\left(\frac{\lambda-\lambda_n}{\sigma}\right)^2} \tag{2}$$

where $\sigma$ is the standard deviation associated with the full width half maximum(FWHM) as $\Delta\lambda = 2\sigma\sqrt{2ln2}$. $\lambda_n$ is the peak wavelength of LED.

Nine LED channels were chosen because good spectral continuity can be ensured by selecting three points in each of the red, green, and blue color bands (380–780 nm). As more points are selected, the final combined spectrum can be made more continuous. The corresponding peak wavelength and FWHM of nine LEDs were summarized in Table 1. The normalised SPDs of nine LEDs were shown in Fig 1.

In Table 1, the wavelengths of 428nm, 450nm, and 480nm, which are more sensitive to the human eye, were adjusted to have the smallest FWHM. Among them, 480nm is beneficial for circadian rhythm regulation. In contrast, 428nm and 450nm, although indispensable for visual processes, are considered somewhat harmful to the human eye. Therefore, their FWHM values were adjusted to be smaller than that of 480nm. The wavelengths of 518nm, 537nm, and 555nm, which fall within the human eye's comfort region, were set to have the largest FWHM values. Meanwhile, the wavelengths of 610nm, 655nm, and 705nm, belonging to the red-light region and being essential for visual processes and beneficial to human skin, were set to have the second-largest FWHM values.

Regarding the visual effect of nine-channel full-spectrum LED color temperature palette, the CRI standard has certain flaws due to its small sample size [31]. In contrast, TM-30 [28], with a larger sample size and considerations for color fidelity and saturation, is regarded as a more rigorous criterion for evaluating color quality. Given that the CRI standard still retains some representativeness, it was combined with the TM-30 standard to comprehensively evaluate the color rendering of the nine-channel LED color temperature palette. Duv and CCT were introduced to assess the chromaticity performance of the light source. Through adjustment of these two parameters, the chromaticity coordinates of the nine-channel LED color temperature palette can be made to approach the blackbody Planck's curve as closely as possible.

**Table 1. Nine LEDs of peak wavelength and FWHM.**

| LED | $\lambda_n$/nm | $\Delta\lambda$/nm |
|---|---|---|
| Purple | 428 | 20 |
| Indigo | 450 | 25 |
| Blue | 480 | 30 |
| Green | 518 | 102 |
| Yellow green-1 | 537 | 102.3 |
| Yellow green-2 | 555 | 114.5 |
| Orange | 610 | 60 |
| Red-1 | 655 | 76 |
| Red-2 | 705 | 84 |

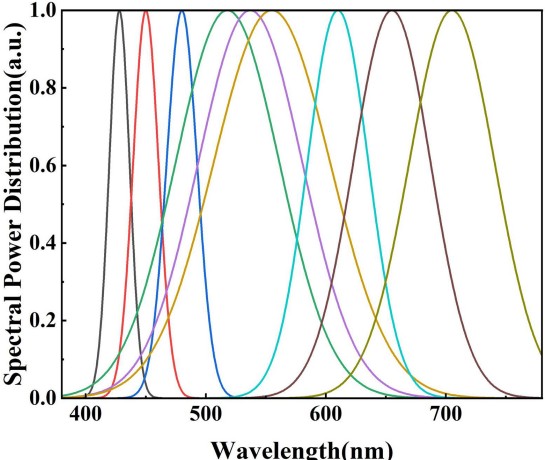

**Fig 1. Normalised SPDs curves of nine LEDs.**

For the non-visual effect of the nine-channel LED color temperature palette, MELR and CAF were optimized based on the maximum and minimum MELR values under the considered Rf constraints and the characteristics of light at different time of the day. A high MELR value is known to inhibit melatonin secretion, while a low MELR value is considered to reduce the inhibitory effect on melatonin secretion. A high CAF value indicates a high concentration of blue-light radiant energy, whereas a low CAF value implies a low blue-light concentration. Therefore, larger MELR and CAF values are required at midday, smaller values are needed at night fall, and medium values are appropriate in the early morning and afternoon.

Next, for the visual and non-visual optimization of the nine-channel LED color temperature palette, corresponding functions were introduced: $F_{CRI}$, $F_{Rf}$, $F_{Rg}$, $F_{Duv}$, $F_{MELR}$ [24], and $F_{CAF}$ [22]. The mathematical expressions of CRI, Rg, and Rf are complex, they can be found in GB/T 5702−2019 Color Rendering Evaluation Method of Light Source and ANSI/IES TM-30–20 (IES 2020), and thus will not be elaborated upon here. The mathematical expressions for the remaining functions are presented as follows:

$$F_{CCT} = 669M^4 - 779M^3 + 3360M^2 - 7047M + 5652, M = \frac{x-0.329}{y-0.187}$$

(3)

$$F_{Duv} = \left[ \frac{|k \cdot u - v + v_m - k \cdot u_m|}{(k^2+1)^{\frac{1}{2}}} \right] sign(v - v_m), k = \frac{v_{m+1}-v_m}{u_{m+1}-u_m} \tag{4}$$

$$F_{MELR} = \frac{\int_{380}^{780} S(\lambda) smel(\lambda) d\lambda}{K_m \int_{380}^{780} S(\lambda) V(\lambda) d\lambda} \tag{5}$$

$$F_{CAF} = \frac{\int_{380}^{780} S(\lambda) C(\lambda) d\lambda}{\int_{380}^{780} S(\lambda) V(\lambda) d\lambda} \tag{6}$$

where (x, y) represents the CIE 1931 chromaticity coordinates of the nine-channel LED color temperature palette, while (u, v) corresponds to its CIE 1960 chromaticity coordinates. The parameter M denotes the reciprocal of the slope of the line of equal correlated color temperature (CCT). $(u_m, v_m)$ is the chromaticity coordinates of the CCT on the Planck black-body curve that is closest to the CCT of the nine-channel LED color temperature palette. $(u_{m+1}, v_{m+1})$ is the adjacent chromaticity coordinates of the chromaticity coordinate $(u_m, v_m)$, and the color temperature between the two color points differs by 10K. And k is the slope of the straight line passing through the two points $(u_m, v_m)$ and $(u_{m+1}, v_{m+1})$. $S(\lambda)$ is the SPD of the nine-channel LED color temperature palette. $smel(\lambda)$ is the melanoptic action spectra. $K_m$ is the maximum spectral luminous efficacy of the radiation for photopic vision($K_m = 683$ lm/W). $C(\lambda)$ is the spectral circadian efficiency. $V(\lambda)$ is the spectral luminous efficiency function for photopic vision.

Firstly, the visual aspect of the nine-channel LED color temperature palette was optimized subject to the following constraints in order to achieve excellent chromaticity and high color rendering of the nine-channel LED color temperature palette:

$$\begin{cases} F_{CCT}(\lambda) \in [2700,6500] \, , |F_{Duv}(\lambda)| < 0.001 \\ F_{CRI}(\lambda) \geq 90, F_{Rf}(\lambda) \geq 90, 90 \leq F_{Rf}(\lambda) \leq 110 \end{cases}$$

Secondly, the non-visual aspect of the nine-channel LED color temperature palette was optimized under the following constraints. $G_{time}$, which represents the desired spectrum at different time of the day, was defined. Based on the maximum and minimum MELR values under the considered Rf constraints and the characteristics of different time of the day, the following requirements were set: maximize MELR and CAF at midday, minimize them at night fall, and set them to intermediate values during the morning/afternoon. Subsequently, the following constraints were introduced:

$$\begin{cases} G_{midday} = MAX\{F_{MELR}(\lambda), F_{CAR}(\lambda)\} \\ G_{night\,fall} = MIN\{F_{MELR}(\lambda), F_{CAR}(\lambda)\} \\ G_{morning/afternoon} = MEDIUM\{F_{MELR}(\lambda), F_{CAR}(\lambda) \\ MELR_{night\,fall} \in [0.4,0.9]; \\ MELR_{morning/afternoon} \in [0.6 ,1.2] \\ MELR_{midday} \in [0.8 ,1.4] \end{cases}$$

Next, the above theoretical model will be simulated to verify its reliability. MATLAB will be employed for the simulation process. The optimization procedure is composed of six steps, as illustrated in Fig 2. To optimize the calculation process and enhance the algorithm's efficiency, the following settings will be adopted in step 1: Based on the Gaussian monochromatic LED model and multi-channel LED mixing technology, nine "steps" (0.01, 0.13, 0.25, 0.38, 0.50, 0.62, 0.74, 0.87, 1) are set within the range of 0–1. The energy ratios of the nine LEDs (L1—L9) are traversed according to the predefined step values. Additionally, for the 2700K—5000K range, the weights of the 555nm, 610nm, 655nm, and 705nm channels are set

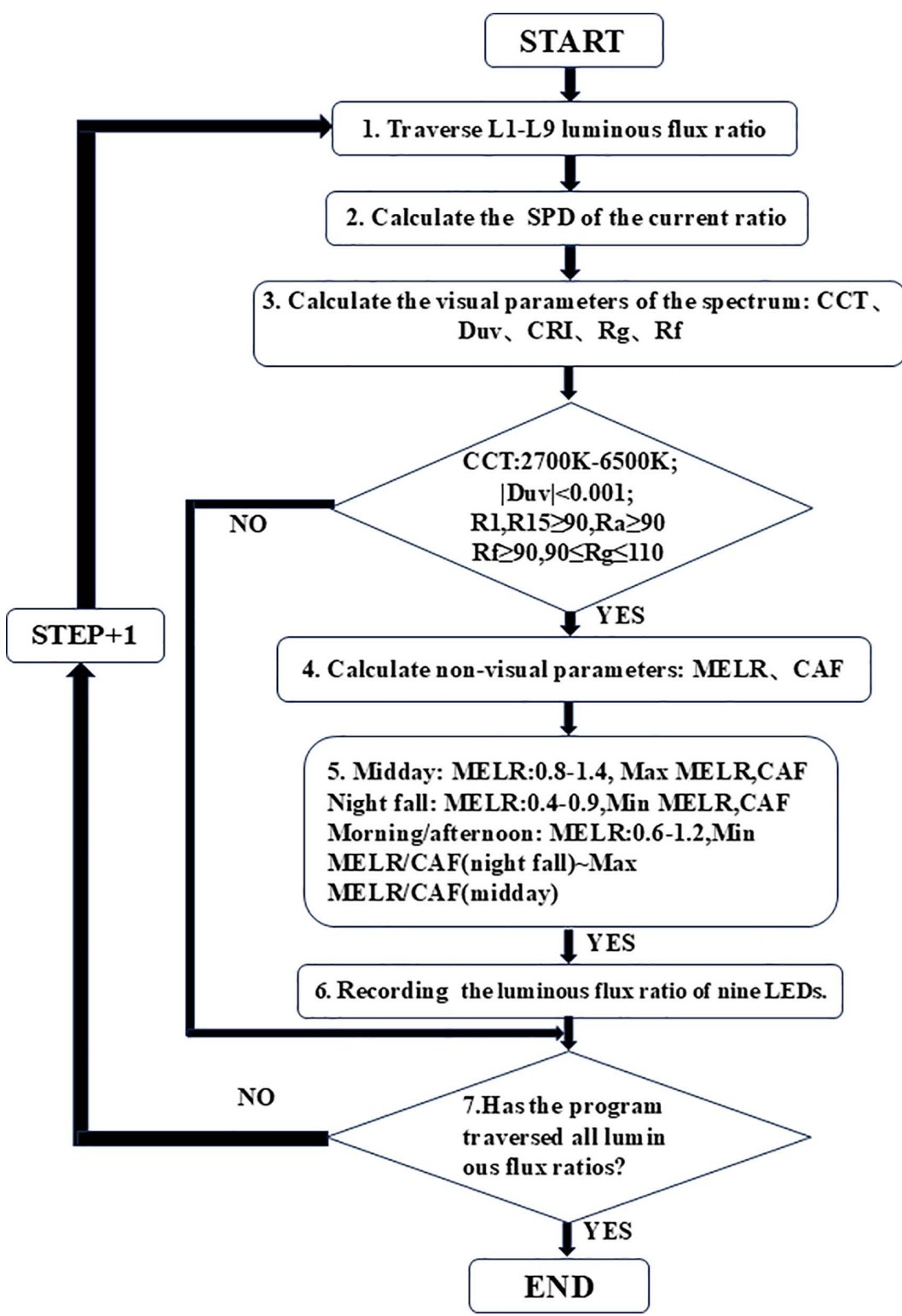

**Fig 2. Spectral optimization flowchart.**

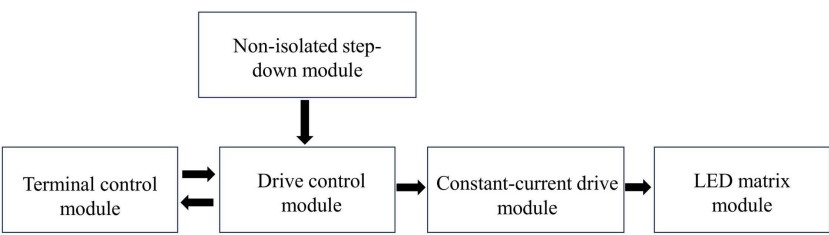

**Fig 3. Hardware design.**

to be 2, 2, 4, and 8 times greater, respectively, whereas for the 5000K—6000K range, the weights of all channels are set to 1.

In the overall hardware design, the driving circuit consists of a terminal control module, a non-isolated step-down module, a drive control module, a constant-current drive module, and a light source module. The connections between the modules are shown in Fig 3. The terminal control module converts the luminous fluxes of the nine LEDs corresponding to the CCTs from 2700K to 6500K into duty cycles and transmits them to the drive control module. The non-isolated step-down module steps down the external 48V power supply to the 3.3V voltage required by the drive control module through a DC-DC step-down chip. The drive control module reads the duty-cycle data from the terminal control module and converts it into PWM(Pulse Width Modulation) signals through a single-chip microcomputer. The constant-current drive module connects the PWM signals output by the single-chip microcomputer to the constant-current drive chip. The constant-current drive chip then outputs currents in different proportions according to different duty cycles. The light source module is composed of the nine LED matrices in Table 1 connected in series or in parallel.

## Result and discussion

Through simulations, the spectrum cluster ranging from 2700K to 6500K was obtained. Generally, considering the characteristics of different time of the day, the CCTs of night fall, morning/afternoon, and midday correspond to the ranges of 2700K—3300K, 3300K—5300K, and 5300K—6500K, respectively.

The luminous flux ratios of the nine LEDs in the spectrum cluster were collected. A set of luminous flux ratios of the nine LEDs was taken as a group, with each group corresponding to a specific CCT. These groups were sorted and

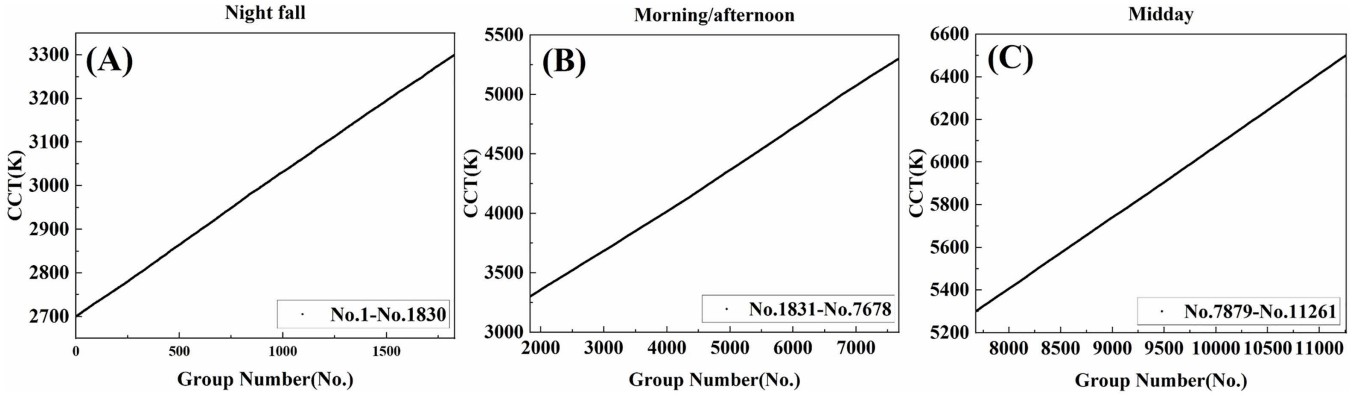

**Fig 4. The CCT curves in different time of the day.** (A) The CCT curve at night fall(group numbers range from 1 to 1830). (B) The CCT curve in the morning/afternoon(group numbers range from 1831 to 7678). (C) The CCT at midday(group numbers range from 7679 to 11261).

numbered in ascending order of CCT, ensuring that each group number was associated with a particular CCT. The CCT curves for different time of the day are shown in Fig 4. The spectrum clusters at night fall have group numbers ranging from 1 to 1830, those in the morning/afternoon have group numbers from 1831 to 7678, and those at midday have group numbers from 7679 to 11261. The CCT can be switched with high precision at 1K interval, resulting in a smooth curve.

In other words, the high-precision smooth CCT switching of the nine-channel LED mentioned can be utilized as a color temperature palette. This allows for the high-precision adjustment to any desired CCT within the range of 2700K—6500K while maintaining high color rendering. Additionally, it ensures that no abrupt visual changes occur during switching, providing a high-quality auxiliary tool for photography, where even a subtle difference in CCT can lead to significantly different visual effects. Similarly, in art exhibitions, where high color rendering and accurate CCT adjustment are required, this nine-channel LED color temperature palettecan be employed to provide a high-quality full-spectrum lighting solution for such fields.

The SPDs of 2700K—6500K are shown in Fig 5A. The three-dimensional SPDs for night fall, morning/afternoon and midday are shown in Fig 5 BCD. Fig 5B shows that the spectrum at night fall have the highest proportion of red light and

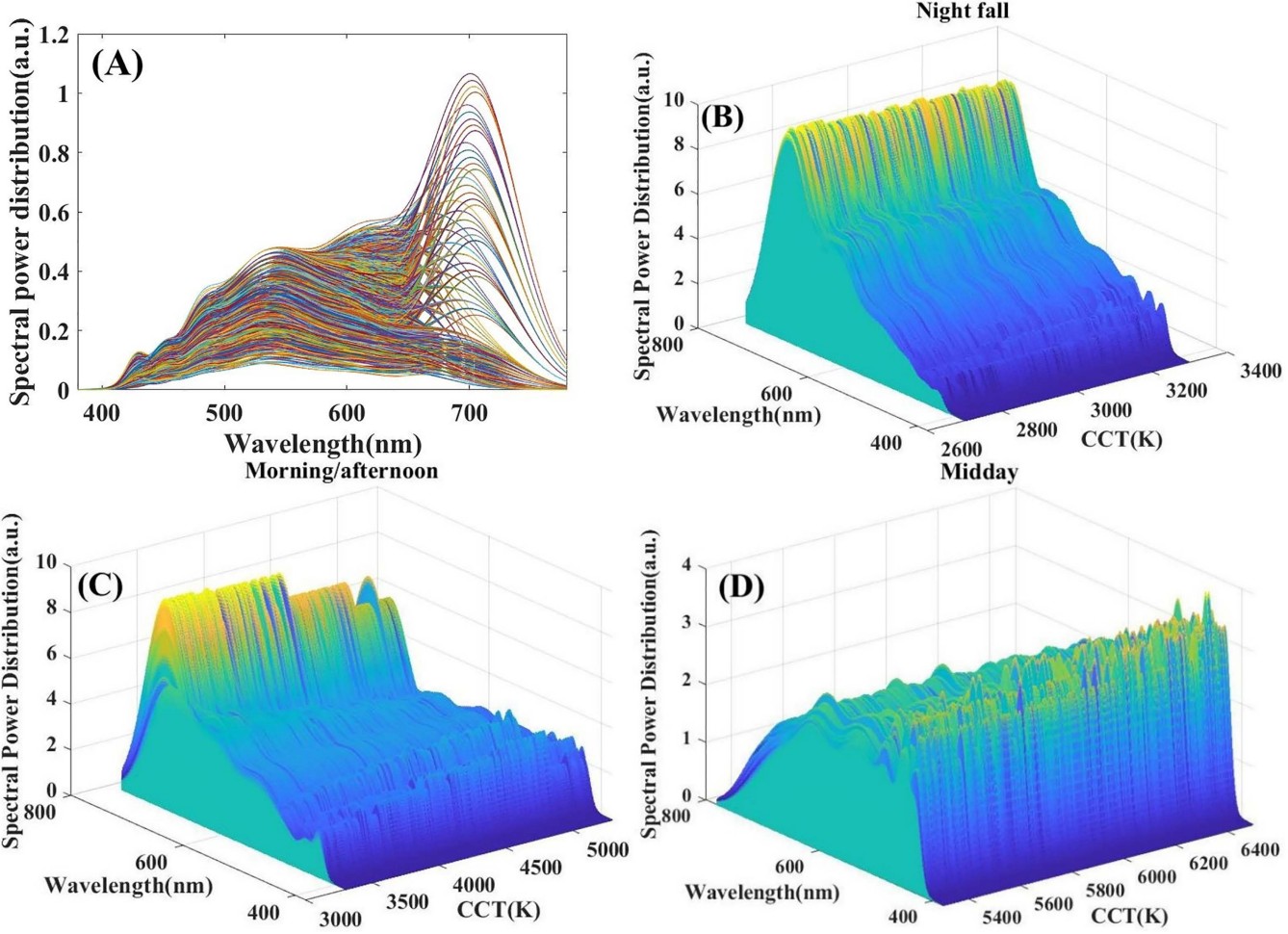

**Fig 5. SPDs in the CCT range of 2700-6500K (from morning to the night fall).** (A) The two-dimensional SPDs from 2700K to 6500K. (B) The three-dimensional SPDs at night fall (2700K—3300K). (C) The three-dimensional SPDs in the morning/afternoon (3300K—5300K). (D) The three-dimensional SPDs at midday (5300K—6500K).

the lowest proportion of blue light. As the CCT increases, the proportion of red light begins to decrease while the proportion of blue light gradually increases, but the increase is small. Fig 5C shows that in the morning/afternoon, compared with the spectrum at night fall, the proportion of red light is significantly reduced and the proportion of blue light is further increased. This is due to the fact that appropriate rhythm stimulation is needed in the morning/afternoon to support people's daily activities. Fig 5D demonstrates that in midday, the proportion of blue light reaches the highest and features stronger spikes. It explains why the non-visual parameters MELR and CAF at midday are set to be higher and at night fall are set to be lower.

Comparing the spectrum shapes of different time of the day, it can be seen that the shape of the spectrum change with different time. At midday the spectrum is mainly concentrated in the blue-green light, while at other times in the yellow-red light. The sensory perceptions provided by the nine-channel LED color temperature palette in different time of the day are as follows: at night fall, users are provided with a comfortable warm white light atmosphere which is closer to the natural light at sunset, allowing people to relax. In the morning/afternoon, users are provided with a softer light and a neutral white light atmosphere which is closer to the natural light in the morning or afternoon, giving people a feeling of serenity. In the midday, users are provided with a cold white light and a transparent, bright, cool atmosphere which is closer to natural midday light, helping people to focus attention. As a result, it can concluded that real-time CCT reproduction of sunlight outdoors can be achieved indoors by this nine-channel LED.

The chromaticity coordinates on the CIE 1931 chromaticity diagram and the Duv for the CCTs ranging from 2700K to 6500K are presented in Fig 6. As seen in Fig 6B, the Duv for the 2700K—6500K range is consistent with |Duv| < 0.001, which is well below the threshold of color difference that can be perceived by the human eye. When combined with the chromaticity coordinates in Fig 6A, it can be concluded that the CCT of the nine-channel LED color temperature palette is very close to the corresponding CCT on Planck's blackbody curve. The trajectory of the CCT is also very similar to Planck's blackbody curve, indicating that excellent chromaticity of the nine-channel LED color temperature palette is achieved.

The CRI values for the 2700K—6500K range are presented in Fig 7. At all CCTs, values of R1—R15 and Ra are maintained above 90. In particular, an average value of 97 for Ra is achieved, and average values of 98, 98, 97, 96, 98, 97, 97, 97, 95, 96, 96, 93, 98, 98, 98 are obtained for R1—R15 respectively. In Fig 7 BC, the values of R9 and R12 of this nine-channel LED color temperature palette are found to exceed 90.

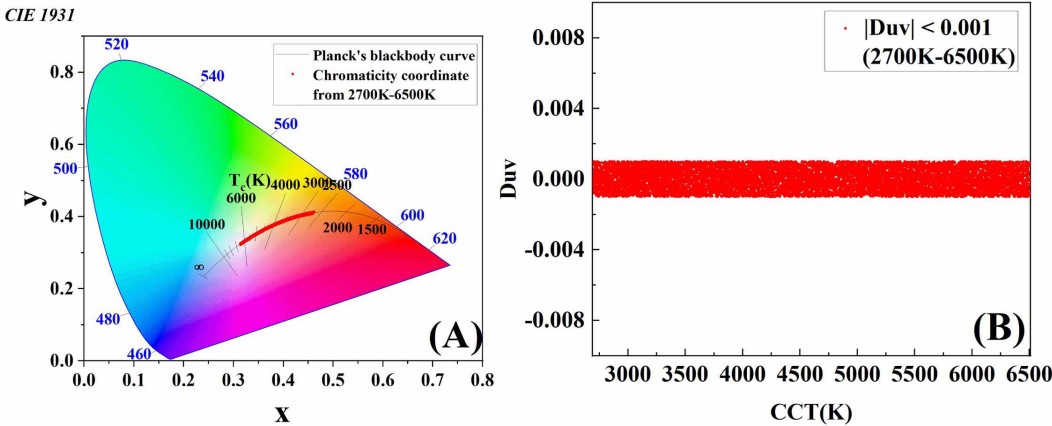

**Fig 6. Chromaticity performance of color points within the CCT range of 2700-6500K. (A) The chromaticity coordinate from 2700K to 6500K on the CIE chromaticity diagram. (B)The Duv from 2700K to 6500K.**

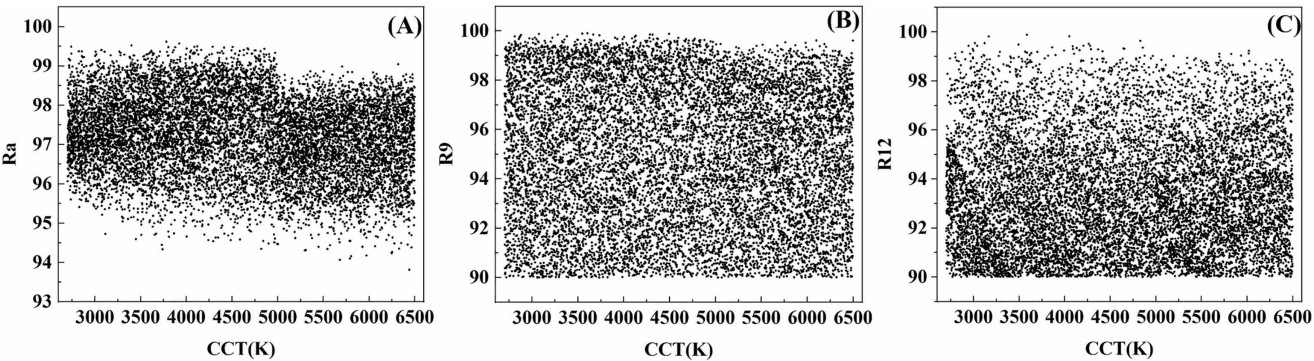

**Fig 7. The CRI from 2700K to 6500K.** (A) The Ra from 2700K to 6500K. (B) The R9(saturated red) from 2700K to 6500K. (C) The R12(saturated blue) from 2700K to 6500K.

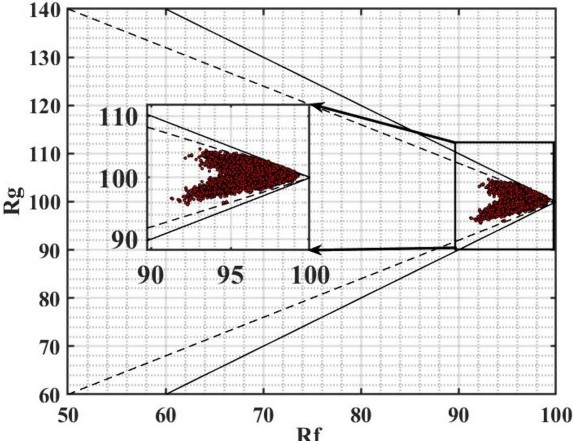

**Fig 8. The Rf and Rg from 2700K to 6500K under the TM-30 standard.** The limit of the Planckian light source is denoted by the dashed line and the limit of the actual light source is denoted by the solid line.

The fidelity index Rf (horizontal coordinate) and the gamut index Rg (vertical coordinate) for the TM-30 color quality evaluation of the nine-channel LED color temperature palette within the 2700K—6500K range are illustrated in Fig 8. As specified in ANSI/IES TM-30–20 [25], Rf and Rg are used to characterize, respectively, the degree of similarity and the change in saturation of each standard color when compared to the reference light source under the test light source. The limit of the Planckian light source is represented by the dashed line, while the limit of the actual light source is represented by the solid line. The fidelity index Rf has a range from 0 to 100. In contrast, the gamut index Rg has no upper limit; however, its value varies with that of Rf, and it is generally considered optimal around 100. Theoretically, their values are distributed within the area enclosed by the dashed line, solid line, and the coordinate axes. As depicted in Fig 8, the Rf and Rg values of the color temperature palette are distributed within the dashed region. High performance is achieved, with the fidelity index reaching Rf ≥ 91 (an average of 96) and the gamut index falling within the range of 96 ≤ Rg ≤ 104 (an average of 101), which indicates that this nine-channel LED color temperature palette exhibits low chromatic aberration and appropriate color saturation.

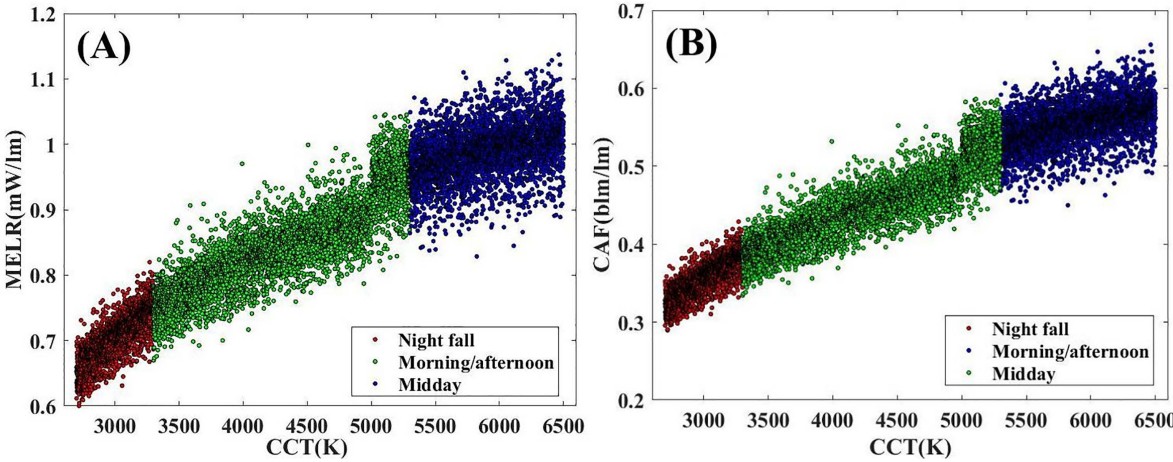

**Fig 9. The non-visual performance of nine-channel LED color temperature palette at different time of the day.** The red marks indicate the data at night fall(2700K-3300K). The green mark indicates the data in the morning/afternoon(3300K-5500K). The blue mark indicates the data at midday(5500K-6500K). (A) The MELR range of nine-channel LED color temperature palette at different time of the day. (B) The CAF range of nine-channel LED color temperature palette at different time of the day.

By integrating the results from the chromaticity diagram in Fig 6 and the CRI values in Fig 7, it can be concluded that this attains excellent color rendering, making it well-suited for applications that demand high color rendering performance.

To further mimic sunlight, a corresponding non-visual effect of this nine-channel LED color temperature palette is also required. The MELR and CAF at different time of the day are presented in Fig 9, where the red, green, and blue parts correspond to night fall, morning/afternoon, and midday, respectively. It is noted that MELR and CAF increase as the CCT is increased. This is because the sensitive wavelengths of MELR and CAF are close to the blue-light region, and the proportion of blue light increases as the CCT rises. As shown in Fig 9, a tunable MELR range of 0.6 mW/lm—1.14 mW/lm and a tunable CAF range of 0.289 blm/lm—0.656 blm/lm are achieved. Both ranges can meet the basic requirements for regulating the circadian rhythm [27]. It can also be observed that the tunable ranges of MELR and CAF vary at different time of the day. Notably, the minimum and maximum values of MELR and CAF at different time of the day increase successively.

It can be concluded that this nine-channel LED color temperature palette can reproduce the CCT while having corresponding non-visual effects, which can better regulate the circadian rhythm. In summary, this nine-channel full-spectrum LED color temperature palette has two modes: in the visual mode, it is a full-spectrum nine-channel LED color temperature palette with continuously tunable CCT; in the non-visual mode, the CCT of outdoor sunlight can be reproduced indoors in real time with corresponding circadian rhythm effects. The multi-functional lighting modes of nine-channel full-spectrum LED color temperature palette are summarized in Table 2.

**Table 2. The multi-functional lighting modes of nine-channel full-spectrum LED color temperature palette.**

| Visual | Ra | Rf | Rg |
|---|---|---|---|
| Full-specturm | 97 | 96 | 101 |
| Non-visual | CCT(K) | MELR(mW/lm) | CAF(blm/lm) |
| Night fall | 2700—3300 | 0.6—0.82 | 0.289—0.429 |
| Morning/afternoon | 3300—5300 | 0.67—1.05 | 0.334—0.586 |
| Midday | 5300—6500 | 0.829—1.14 | 0.450—0.656 |

**Table 3. Comparison of key parameters of four-channel, five-channel, six-channel and nine-channel LEDs.**

| | Four-channel LED | Five-channel LED | Six-channel LED | Nine-channel LED |
|---|---|---|---|---|
| CCT(K) | Product-showcase: 2971–6568(513K) | 5:30am-12:03 pm:3638–5737 | Work: 3552–6633(616K) | Night fall: 2700–3300(1K) |
| | Office-at-night: 3018−6608(513K) | (150K) 12:30 pm-19:04 | Leisure: 1783–3921(428K) | Morning/afternoon: 3300-5300(1K) |
| | Bedroom-at-night: 2991–6670(525K) | pm:3554–5694 (151K) | Daytime indoor: 2792-5655(477K) | Midday: 5300–6500(1K) |
| Ra | Product-showcase: 95 | | Work: 85 | Night fall: 97 |
| | Office-at-night: 81 | 5:30am-12:03 pm: 93 | Leisure: 84 | Morning/afternoon: 97 |
| | Bedroom-at-night: 89 | 12:30 pm-19:04 pm: 92 | Daytime indoor:97 | Midday: 97 |
| Duv | \|Duv\|<0.005 | \|Duv\|<0.004 | \|Duv\|<0.0039 | \|Duv\|<0.001 |
| $R_g$ | – | – | – | 101 |
| $R_f$ | – | – | – | 96 |
| MELR(mw/lm) | – | – | – | Night fall: 0.6–0.82 |
| | | | | Morning/afternoon: 0.67–1.05 |
| | | | | Midday: 0.829–1.14 |
| CAF(blm/lm) | Product-showcase: 0.426–0.887 | | Work: 0.530–1.022 | Night fall: 0.289–0.429 |
| | Office-at-night: 0.453–0.934 | 5:30am-12:03 pm: 0.548–0.837 | Leisure: 0.097–0.471 | Morning/afternoon: 0.334–0.586 |
| | Bedroom-at-night: 0.410–0.881 | 12:30 pm-19:04 pm: 0.525–0.834 | Daytime indoor:- | Midday: 0.450–0.656 |

As shown in Table 3, the key parameters of the nine-channel LED color temperature palette developed in this study are compared with those of four-channel [21], five-channel [25], and six-channel LEDs [22]. The average adjustment accuracy of the correlated color temperature (CCT) is indicated in parentheses following the CCT adjustment range in the table. A dash ("−") denotes that the parameter was not addressed in the respective study or scenario.

The four-, five-, and six-channel LEDs demonstrated relatively low modulation accuracy in CCT adjustment, whereas the nine-channel palette achieved high precision, with an accuracy of 1 K. In terms of color rendering, the palette consistently outperformed the other configurations, delivering a higher Color Rendering Index (CRI) across all scenarios. The use of the TM-30 standard (Rf, Rg) further confirmed that the palette met high color rendering requirements. Additionally, the palette exhibited a lower Duv value compared to the other LEDs, indicating superior chromaticity. Regarding non-visual effects, the palette achieved a lower Circadian Action Factor (CAF) in low-CCT scenarios, suggesting reduced blue light stimulation—making it particularly suitable for relaxation environments. Furthermore, the Melanopic Equivalent Daylight (MELR) performance of the nine-channel palette met the essential requirements previously discussed.

In developing the multi-channel LED model, an ideal Gaussian distribution function was initially employed to simulate the spectral power distribution (SPD) of monochromatic LEDs. However, this approach does not account for the inherent asymmetry observed in real LEDs, potentially limiting the applicability of the simulation results in practical scenarios. To address this, an asymmetry factor was introduced into the Gaussian distribution function to evaluate its impact on simulation outcomes.

The asymmetric Gaussian distribution was constructed by dividing the symmetric function into left and right halves at the central wavelength, introducing a 5% difference in area between the two sides. In Table 4, the first column (Group 1) represents a case where the left-half area is 5% larger than the right, while the second column (Group 2) shows the opposite, with the right-half area 5% larger than the left. The third column serves as the control group, representing the ideal symmetric model. The results indicate that a color temperature modulation accuracy of 1 K can still be maintained even with the introduction of asymmetry. In terms of color rendering, both Ra and Rf remained unchanged in Groups 1 and 2. Rg remained stable in Group 1 but showed a slight decrease of approximately 1% in Group 2 compared to the control

**Table 4. The key parameter values after adjusting the symmetry of the Gaussian distribution function.**

| | Area on left side 5% larger than that on the right side | Area on right side 5% larger than that on the left side | Symmetrical |
|---|---|---|---|
| CCT(K) | Night fall: 2700–3300(1K) | Night fall: 2700–3300(1K) | Night fall:2700–3300(1K) |
| | Morning/afternoon: 3300–5300(1K) | Morning/afternoon: 3300–5300(1K) | Morning/afternoon:3300–5300(1K) |
| | Midday: 5300–6500(1K) | Midday: 5300–6500(1K) | Midday:5300–6500(1K) |
| Ra | 97 | 97 | 97 |
| Duv | |Duv|<0.001 | |Duv|<0.001 | |Duv|<0.001 |
| $R_g$ | 101 | 100 | 101 |
| $R_f$ | 96 | 96 | 96 |
| MELR(mw/lm) | Night fall: 0.611–0.859; | Night fall: 0.572–0.827 | Night fall: 0.6–0.82 |
| | Morning/afternoon: 0.646–1.06 | Morning/afternoon: 0.630–1.02 | Morning/afternoon: 0.67–1.05 |
| | Midday: 0.846–1.17 | Midday: 0.794–1.12 | Midday: 0.829–1.14 |
| CAF(blm/lm) | Night fall: 0.297–0.457; | Night fall: 0.274–0.432; | Night fall: 0.289–0.429 |
| | Morning/afternoon: 0.343–0.622 | Morning/afternoon: 0.321–0.592 | Morning/afternoon: 0.334–0.586 |
| | Midday: 0.461–0.680 | Midday: 0.429–0.645 | Midday: 0.450–0.656 |

group. For non-visual parameters, the absolute values of the boundary value change ratios ranged from a minimum of 0.7% to a maximum of 6.53%.

Additionally, it is important to analyze the impact of variations in the full width at half maximum (FWHM) of LEDs on the simulation results. This analysis contributes to a deeper understanding of the stability and robustness of the nine-channel LED color temperature palette in practical applications.

Table 5 presents the influence on key parameters when the FWHM of the LEDs is adjusted by ±10%. The third column (Control Group) represents the baseline FWHM values used in this study. The first column (Group 1) corresponds to a scenario where the FWHM is reduced by 5% relative to the control group, while the second column (Group 2) reflects a 5% increase. As shown in Table 5, a color temperature modulation accuracy of 1K is maintained even when the FWHM is varied. In terms of color rendering, both Ra and Rf values remained unchanged in Groups 1 and 2 compared to the control group. The Rg value remained stable in Group 1 but decreased by approximately 1% in Group 2. For non-visual parameters, the absolute values of the boundary value change ratios ranged from a minimum of 0.72% to a maximum of 12.29%.

**Table 5. The key parameter values after adjusting the FWHM of the LEDs by ±10%.**

| | Lower 10% FWHM | Higher 10% FWHM | Original FWHM |
|---|---|---|---|
| CCT(K) | Night fall: 2700–3300(1K) | Night fall: 2700–3300(1K) | Night fall: 2700–3300(1K) |
| | Morning/afternoon: 3300–5300(1K) | Morning/afternoon: 3300–5300(1K) | Morning/afternoon: 3300–5300(1K) |
| | Midday: 5300–6500(1K) | Midday: 5300–6500(1K) | Midday: 5300–6500(1K) |
| Ra | 97 | 97 | 97 |
| Duv | |Duv|<0.001 | |Duv|<0.001 | |Duv|<0.001 |
| $R_g$ | 101 | 100 | 101 |
| $R_f$ | 96 | 96 | 96 |
| MELR(mw/lm) | Night fall: 0.564–0.809 | Night fall: 0.615–0.872 | Night fall: 0.6–0.82 |
| | Morning/afternoon: 0.630–1.02 | Morning/afternoon: 0.687–1.13 | Morning/afternoon: 0.67–1.05 |
| | Midday: 0.793–1.11 | Midday: 0.835–1.18 | Midday: 0.829–1.14 |
| CAF(blm/lm) | Night fall: 0.264–0.414 | Night fall: 0.312–0.444 | Night fall: 0.289–0.429 |
| | Morning/afternoon: 0.305–0.560 | Morning/afternoon: 0.353–0.658 | Morning/afternoon: 0.334–0.586 |
| | Midday: 0.420–0.621 | Midday: 0.462–0.699 | Midday: 0.450–0.656 |

In summary, when the Gaussian distribution function includes a 5% asymmetry and the FWHM is adjusted by ±10%, the resulting variations in simulation outcomes remain within an acceptable range. These findings indicate that the nine-channel LED color temperature palette model, based on the Gaussian distribution function, demonstrates a high degree of stability and reliability for practical applications.

## Conclusion

In this paper, a multi-functional nine-channel full-spectrum LED color temperature palette is proposed. It features continuously tunable CCT at 1K interval within the range of 2500K—6500K, high color rendering with Ra > 94, R1—R15 > 90, Rf > 91, and 96 ≤ Rg ≤ 104. A |Duv| < 0.001 is also achieved. The circadian effect is also realized, with MELR ranging from 0.6 mW/lm to 1.14 mW/lm and CAF ranging from 0.289 blm/lm to 0.656 blm/lm. The high-precision smooth CCT switching of the nine-channel full-spectrum LED can be utilized as a color temperature palette. This enables the desired CCT to be adjusted to, and the CCT of sunlight to be reproduced with high precision and high color rendering, while also having corresponding non-visual effects. To simulate non-ideal practical conditions and assess the model's robustness, the Gaussian distribution function of the monochromatic LED spectrum is adjusted to yield 5% asymmetry in integral areas, and the FWHM is offset by ±10%. Through these adjustments, the model is shown to retain good stability, demonstrating its reliability under varied conditions. The multi-functional nine-channel LED full-spectrum color temperature palette developed in this study is of positive significance for fields requiring high color rendering and circadian rhythm regulation. It paves the way for the development of intelligent and tunable high-quality full-spectrum light sources.

## Supporting information

**S1. Data access.**
(DOCX)

## Acknowledgments

The authors thank the reviewers for their valuable suggestions regarding this work.

## Author contributions

**Data curation:** Jingjing Liang.

**Methodology:** Jingjing Liang, Hui Cao.

**Supervision:** Hui Cao, Donghua Xu, Xiaobing Lu, Feijiang Huang, Javid Atai.

**Validation:** Hui Cao.

**Writing – original draft:** Jingjing Liang.

**Writing – review & editing:** Hui Cao, Donghua Xu, Xiaobing Lu, Javid Atai.

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
