## [Decision Letter · Decision Letter 0]

21 Apr 2025

PONE-D-25-14407A multi-functional nine channels full-spectrum light emitting diodes color temperature palettePLOS ONE

Dear Dr. Cao,

Thank you for submitting your manuscript to PLOS ONE. After careful consideration, we feel that it has merit but does not fully meet PLOS ONE’s publication criteria as it currently stands. Therefore, we invite you to submit a revised version of the manuscript that addresses the points raised during the review process.

We look forward to receiving your revised manuscript.

Kind regards,

Amitava Mukherjee, ME, Ph.D.

Academic Editor

PLOS ONE

“Bureau of Education of Guangzhou Municipality(202234641); Department of Education of Guangdong Province(2020ZDZX3072,2021KTSCX098); Research Project of Construction National Science and Technology Think Tank of Guangdong Province (SXK20220201035); Education and Science Program of Guangdong Province (2022GXJK308); Project of Construction Discipline Scientific Research Capability Improvement of Guangdong Province(2022ZDJS098, 2022ZDJS096); Guangzhou Key R&D Program Agriculture and Social Development Science and Technology Project (202206010034) and Guangzhou Municipal Basic Research Project jointly funded by municipal schools (institutes) (2023A03J0834).”

4. In the online submission form, you indicated that [Data underlying the results presented in this paper are not publicly available at this time but may be obtained from the authors upon reasonable request.].

6. PLOS requires an ORCID iD for the corresponding author in Editorial Manager on papers submitted after December 6th, 2016. Please ensure that you have an ORCID iD and that it is validated in Editorial Manager. To do this, go to ‘Update my Information’ (in the upper left-hand corner of the main menu), and click on the Fetch/Validate link next to the ORCID field. This will take you to the ORCID site and allow you to create a new iD or authenticate a pre-existing iD in Editorial Manager.

Reviewers' comments:

Reviewer's Responses to Questions

**Comments to the Author**

1. Is the manuscript technically sound, and do the data support the conclusions?

Reviewer #1: Yes

2. Has the statistical analysis been performed appropriately and rigorously? 

Reviewer #1: Yes

3. Have the authors made all data underlying the findings in their manuscript fully available?

Reviewer #1: Yes

4. Is the manuscript presented in an intelligible fashion and written in standard English?

Reviewer #1: Yes

5. Review Comments to the Author

Reviewer #1: This manuscript presents a detailed theoretical framework and simulation for designing a multi-functional nine-channel full-spectrum LED system capable of 1K-interval tunable correlated color temperature (CCT) across 2500K–6500K. The system features high color rendering (Ra > 94, R1–R15 > 90, Rf > 91) and incorporates parameters addressing circadian effects (MELR and CAF). The work is timely and relevant to the development of human-centric lighting, and the authors have made a commendable effort in exploring both visual and non-visual metrics through simulation. However, before the manuscript can be accepted for publication, the following issues need to be addressed:

1.The manuscript relies solely on MATLAB-based simulation. No experimental or hardware-based demonstration is presented to validate the proposed model or assess real-world implementation challenges such as thermal management, control electronics, or spectrum stability. The inclusion of a prototype or even preliminary experimental data would greatly strengthen the manuscript.

2. Although the authors briefly mention other multi-channel approaches, they do not provide a direct quantitative comparison. Including a comparative table summarizing key metrics (Ra, Rf, Rg, Duv, MELR, CAF) versing 4-, 5-, or 6-channel designs would clarify the specific advantages and performance enhancements of the nine-channel configuration.

3. The LED spectra are modeled using ideal Gaussian distributions. This simplification may not reflect the asymmetrical and secondary peaks seen in real LEDs, particularly phosphor-converted devices. The authors should discuss the limitations of this assumption and its impact on simulation of fidelity.

4. There is no evaluation of how robust the simulation results are to variations in LED flux ratios or FWHM values. A sensitivity analysis on these parameters would provide insights into the stability of the optimized performance under practical variations.

5. The manuscript would benefit from language polishing. There are grammatical errors, awkward phrasing, and redundancy throughout the text. For example, phrases such as "serve as a color temperature palette" and "the values...are attained" should be revised for correctness and clarity.

6. Several figures lack sufficient clarity, proper labeling, and detailed captions. Figures 4 through 8, in particular, need improved resolution and annotations to aid reader comprehension. Axis labels, units, legends, and figure captions should be made more informative.

7. Define all abbreviations at first use and maintain consistency (e.g., SPD, MELR, CAF).

8. Include discussion on control hardware and how 1K CCT tuning precision is practically achieved.

9. Reformat equations for clarity, ensuring all symbols and parameters are defined within the text.

10. To enhance the scholarly depth of the manuscript and align with recent advancements in tunable and circadian LED systems, the authors are strongly encouraged to cite the following relevant studies: OSA Continuum, 2019, Vol. 2, Issue 8, pp. 2413–2427. DOI: 10.1364/OSAC.2.002413 and Near-Unity PLQY of Cs₃Cu₂X₅ (X = Cl, Br) for High-Efficiency White Light-Emitting Diodes with Exceptional Color Quality, Advanced Materials, 2025, Article No. 2500083. DOI: 10.1002/adma.202500083. These references will support the discussion of tunable spectrum design and reinforce the context for high-quality full-spectrum LED development.

6. PLOS authors have the option to publish the peer review history of their article (what does this mean? ). If published, this will include your full peer review and any attached files.

**Do you want your identity to be public for this peer review?** For information about this choice, including consent withdrawal, please see our Privacy Policy .

Reviewer #1: No

---

## [Author Response · Author response to Decision Letter 1]

29 May 2025

To reviewer:

I have addressed the reviewers' comments in cover letter.

In case you can't see it, the content is as follows:

Dear reviewer ,

Thanks for the constructive comments.

In the revised version, we have made some changes in line with the reviewer's remarks. A summary of the changes and our response to the reviewer's comments is appended below.

1.The manuscript relies solely on MATLAB-based simulation. No experimental or hardware-based demonstration is presented to validate the proposed model or assess real-world implementation challenges such as thermal management, control electronics, or spectrum stability. The inclusion of a prototype or even preliminary experimental data would greatly strengthen the manuscript.

Response: The focus of this article is on the construction of a theoretical model and its verification using simulations. The aim is to fully explore, optimize, and verify the core principles and algorithms of the proposed model through MATLAB simulations, thereby ensuring the validity and rationality of the theoretical model. This is helpful in reducing the trial-and-error cost and improves the efficiency during the subsequent experimental stage. In this paper, we have demonstrated certain results at the theoretical simulation level and completed the optimization of key parameters and the verification of the algorithm's feasibility. Currently, we are working on the hardware construction. We plan to build the corresponding experimental platform to test the theoretical model through practical tests and further improve the system.

2.Although the authors briefly mention other multi-channel approaches, they do not provide a direct quantitative comparison. Including a comparative table summarizing key metrics (Ra, Rf, Rg, Duv, MELR, CAF) versing 4-, 5-, or 6-channel designs would clarify the specific advantages and performance enhancements of the nine-channel configuration.

Response: To address the comments of the reviewer, in the revised version, we have added Table 3 and a detailed discussion. As shown in Table 3, the key parameters of the nine-channel LED color temperature palette developed in this study are compared with those of four-channel [21], five-channel [25], and six-channel LEDs [22]. The average adjustment accuracy of the correlated color temperature (CCT) is indicated in parentheses following the CCT adjustment range in the table. A dash ("–") denotes that the parameter was not addressed in the respective study or scenario.

The four-, five-, and six-channel LEDs demonstrated relatively low modulation accuracy in CCT adjustment, whereas the nine-channel palette achieved high precision, with an accuracy of 1 K. In terms of color rendering, the palette consistently outperformed the other configurations, delivering a higher Color Rendering Index (CRI) across all scenarios. The use of the TM-30 standard (Rf, Rg) further confirmed that the palette met high color rendering requirements. Additionally, the palette exhibited a lower Duv value compared to the other LEDs, indicating superior chromaticity. Regarding non-visual effects, the palette achieved a lower Circadian Action Factor (CAF) in low-CCT scenarios, suggesting reduced blue light stimulation—making it particularly suitable for relaxation environments. Furthermore, the Melanopic Equivalent Daylight (MELR) performance of the nine-channel palette met the essential requirements previously discussed.

Table 3. Comparison of key parameters of four-channel, five-channel, six-channel and nine-channel LEDs

3.The LED spectra are modeled using ideal Gaussian distributions. This simplification may not reflect the asymmetrical and secondary peaks seen in real LEDs, particularly phosphor-converted devices. The authors should discuss the limitations of this assumption and its impact on simulation of fidelity.

Response: To address the comments of the reviewer, in the revised version, we have added Table 4 and a detailed discussion. In developing the multi-channel LED model, an ideal Gaussian distribution function was initially employed to simulate the spectral power distribution (SPD) of monochromatic LEDs. However, this approach does not account for the inherent asymmetry observed in real LEDs, potentially limiting the applicability of the simulation results in practical scenarios. To address this, an asymmetry factor was introduced into the Gaussian distribution function to evaluate its impact on simulation outcomes.

The asymmetric Gaussian distribution was constructed by dividing the symmetric function into left and right halves at the central wavelength, introducing a 5% difference in area between the two sides. In Table 4, the first column (Group 1) represents a case where the left-half area is 5% larger than the right, while the second column (Group 2) shows the opposite, with the right-half area 5% larger than the left. The third column serves as the control group, representing the ideal symmetric model. The results indicate that a color temperature modulation accuracy of 1 K can still be maintained even with the introduction of asymmetry. In terms of color rendering, both Ra and Rf remained unchanged in Groups 1 and 2. Rg remained stable in Group 1 but showed a slight decrease of approximately 1% in Group 2 compared to the control group. For non-visual parameters, the absolute values of the boundary value change ratios ranged from a minimum of 0.7% to a maximum of 6.53%.

The following is the detailed calculation process (not included in the manuscript): When constructing the asymmetric Gaussian distribution function, we divide the function into left and right halves with the central wavelength as the dividing line, and reflect the asymmetry according to the different areas of the two sides. Here, we set the area difference between the two sides to be 5%. The first column in the table shows that the area of the left part of the Gaussian distribution function is 5% more than that of the right part (Group 1), the second column shows that the area of the right part of the Gaussian distribution function is 5% more than that of the left part (Group 2), and the third column is the ideal Gaussian distribution model (control group). It can be seen from Table 4 that when the Gaussian distribution function is asymmetric, the color temperature can still achieve a modulation accuracy of 1K. In terms of color rendering, the Ra and Rf indicators remain unchanged; the Rg indicator remains unchanged in Group 1 and decreases by about -1% in Group 2 compared with the control group. Among the non-visual effect indicators, we compare the change ratios of the minimum and maximum boundary values.

Group 1:

Night fall period: The change ratio of the upper boundary of MELR is 4.76%, and the change ratio of the lower boundary is 1.83%.

Morning/afternoon period: The change ratio of the upper boundary of MELR is 0.95%, and the change ratio of the lower boundary is -3.58%.

Midday period: The change ratio of the upper boundary of MELR is 2.63%, and the change ratio of the lower boundary is 2.05%.

Similarly:

Night fall period: The change ratio of the upper boundary of CAF is 6.53%, and the change ratio of the lower boundary is 2.77%.

Morning/afternoon period: The change ratio of the upper boundary of CAF is 6.14%, and the change ratio of the lower boundary is 2.69%.

Midday period: The change ratio of the upper boundary of CAF is 3.66%, and the change ratio of the lower boundary is 2.44%.

Group 2:

Night fall period: The change ratio of the upper boundary of MELR is 0.85%, and the change ratio of the lower boundary is -4.67%.

Morning/afternoon period: The change ratio of the upper boundary of MELR is -2.86%, and the change ratio of the lower boundary is -5.97%.

Midday period: The change ratio of the upper boundary of MELR is -1.75%, and the change ratio of the lower boundary is -4.22%.

Similarly:

Night fall period: The change ratio of the upper boundary of CAF is 0.7%, and the change ratio of the lower boundary is -5.19%.

Morning/afternoon period: The change ratio of the upper boundary of CAF is 1.02%, and the change ratio of the lower boundary is -3.89%.

Midday period: The change ratio of the upper boundary of CAF is -1.68%, and the change ratio of the lower boundary is -4.67%.

Table 4. The key parameter values after adjusting the symmetry of the Gaussian distribution function.

Area on left side 5% larger than that on the right side Area on right side 5% larger than that on the left side Symmetrical

4.There is no evaluation of how robust the simulation results are to variations in LED flux ratios or FWHM values. A sensitivity analysis on these parameters would provide insights into the stability of the optimized performance under practical variations.

Response: To address the comments of the reviewer, in the revised version, we have added Table 5 and a detailed discussion. Also, we have analyzed the impact of changes in the FWHM of LEDs on the simulation results. This analysis contributes to a deeper understanding of the stability and robustness of the nine-channel LED color temperature palette in practical applications.

Table 5 presents the influence on key parameters when the FWHM of the LEDs is adjusted by ±10%. The third column (Control Group) represents the baseline FWHM values used in this study. The first column (Group 1) corresponds to a scenario where the FWHM is reduced by 5% relative to the control group, while the second column (Group 2) reflects a 5% increase. As shown in Table 5, a color temperature modulation accuracy of 1 K is maintained even when the FWHM is varied. In terms of color rendering, both Ra and Rf values remained unchanged in Groups 1 and 2 compared to the control group. The Rg value remained stable in Group 1 but decreased by approximately 1% in Group 2. For non-visual parameters, the absolute values of the boundary value change ratios ranged from a minimum of 0.72% to a maximum of 12.29%.

The following is the detailed calculation process(not included in the manuscript): The third column of the table shows the full width at half maximum (FWHM) in the original text, serving as the control group. The first column indicates that the LED's FWHM is 5% smaller than that of the control group, and the second column shows that the LED's FWHM is 5% larger than that of the control group. As can be seen from Table 1, when the FWHM of the lamp bead is adjusted by ±10%, the color temperature can still reach a modulation accuracy of 1K. Regarding color rendering, compared with the control group, the Ra and Rf values of the first and second groups remain unchanged. The Rg value of the first group remains the same, and the Rg value of the second group decreases by about 1%. For the non-visual effect indicators, we compared the percentage changes of the minimum and maximum boundary values.

Group 1:

During the Night Fall period, the percentage change of the upper boundary of MELR is -1.34%, and that of the lower boundary is -6%.

During the morning/afternoon period, the percentage change of the upper boundary of MELR is -2.86%, and that of the lower boundary is -5.97%.

During the midday period, the percentage change of the upper boundary of MELR is -2.63%, and that of the lower boundary is -4.34%.

Similarly, during the Night Fall period, the percentage change of the upper boundary of CAF is -3.49%, and that of the lower boundary is -8.65%.

During the morning/afternoon period, the percentage change of the upper boundary of CAF is -4.44%, and that of the lower boundary is -8.68%.

During the midday period, the percentage change of the upper boundary of CAF is -5.34%, and that of the lower boundary is -6.67%.

Group 2:

During the Night Fall period, the percentage change of the upper boundary of MELR is 6.34%, and that of the lower boundary is 2.5%.

During the morning/afternoon period, the percentage change of the upper boundary of MELR is 7.62%, and that of the lower boundary is 2.54%.

During the midday period, the percentage change of the upper boundary of MELR is 3.51%, and that of the lower boundary is 0.72%.

Similarly, during the Night Fall period, the percentage change of the upper boundary of CAF is 3.49%, and that of the lower boundary is 7.96%.

During the morning/afternoon period, the percentage change of the upper boundary of CAF is 12.29%, and that of the lower boundary is 5.69%.

During the midday period, the percentage change of the upper boundary of CAF is 6.55%, and that of the lower boundary is 2.67%.

Table 5. The key parameter values after adjusting the FWHM of the LEDs by ±10%.

5.The manuscript would benefit from language polishing. There are grammatical errors, awkward phrasing, and redundancy throughout the text. For example, phrases such as "serve as a color temperature palette" and "the values...are attained" should be revised for correctness and clarity.

Response: We have polished the language of the article.

6.Several figures lack sufficient clarity, proper labeling, and detailed captions. Figures 4 through 8, in particular, need improved resolution and annotations to aid reader comprehension. Axis labels, units, legends, and figure captions should be made more informative.

Response: The labels of each figure have been corrected to correspond with the figure captions. To improve the clarity, the captions of Figures 4, 5, 8, and 9 have been rewritten, and Figures 4 and 6 have been properly annotated. Regarding the image clarity, the images were clear when I uploaded them to the system, but they became unclear after the platform integrated them into a PDF file. However, they are clear again after downloading.

7.Define all abbreviations at first use and maintain consistency (e.g., SPD, MELR, CAF).

Response: The abbreviations have been defined when they are used for the first time and consistency has been maintained.

8.Include discussion on control hardware and how 1K CCT tuning precision is practically achieved.

Response: To address the comments of the reviewer, in the revised version, we have added Figure 3 and a detailed discussion. In the overall hardware design, the driving circuit consists of a terminal control module, a non-isolated step-down module, a drive control module, a constant-current drive module, and a light source module. The connections between the modules are shown in Fig 3. The terminal control module converts the luminous fluxes of the nine LEDs corresponding to the CCTs from 2700K to 6500K into duty cycles and transmits them to the drive control module. The non-isolated step-down module steps down the external 48V power supply to the 3.3V voltage required by the drive control module through a DC-DC step-down chip. The drive control module reads the duty-cycle data from the terminal control module and converts it into PWM(Pulse Width Modulation) signals through a single-chip microcomputer. The constant-current drive module connects the PWM signals output by the single-chip microcomputer to the constant-current drive chip. The constant-current drive chip then outputs currents in different proportions according to different duty cycles. The light source module is composed of the nine LED matrices in Table 1 connected in series or in parallel.

Fig 3. Hardware design.

9.Reformat equations for clarity, ensuring all symbols and parameters are defined within the text.

Response: The formulas have been reformatted, and it has been ensured that all the symbols and parameters are defined in the text.

10.To enhance the scholarly depth of the manuscript and align with recent advancements in tunable and circadian LED systems, the authors are strongly encouraged to cite the following relevant studies: OSA Continuum, 2019, Vol. 2, Issue 8, pp. 2413–2427. DOI: 10.1364/OSAC.2.002413 and Near-Unity PLQY of Cs₃Cu₂X₅ (X = Cl, Br) for High-Efficiency White Light-Emitting Diodes with Exceptional Color Quality, Advanced Materials, 2025, Article No. 2500083. DOI: 10.1002/adma.202500083. These references will support the discussion of tunable spectrum design and reinforce the context for high-quality full-spectrum LED development.

Response: In the revised version, we have cited these two articles. The section of the manuscript where these papers are

---

## [Decision Letter · Decision Letter 1]

20 Jun 2025

A multi-functional nine channels full-spectrum light emitting diodes color temperature palette

PONE-D-25-14407R1

Dear Dr. Cao,

We’re pleased to inform you that your manuscript has been judged scientifically suitable for publication and will be formally accepted for publication once it meets all outstanding technical requirements.

Kind regards,

Amitava Mukherjee, ME, Ph.D.

Academic Editor

PLOS ONE

Additional Editor Comments (optional):

Reviewers' comments:

Reviewer's Responses to Questions

**Comments to the Author**

1. If the authors have adequately addressed your comments raised in a previous round of review and you feel that this manuscript is now acceptable for publication, you may indicate that here to bypass the “Comments to the Author” section, enter your conflict of interest statement in the “Confidential to Editor” section, and submit your "Accept" recommendation.

Reviewer #1: All comments have been addressed

2. Is the manuscript technically sound, and do the data support the conclusions?

Reviewer #1: Yes

3. Has the statistical analysis been performed appropriately and rigorously? 

Reviewer #1: Yes

4. Have the authors made all data underlying the findings in their manuscript fully available?

Reviewer #1: Yes

5. Is the manuscript presented in an intelligible fashion and written in standard English?

Reviewer #1: Yes

6. Review Comments to the Author

Reviewer #1: The authors have addressed all the comments and suggestions. The current version can be accepted as it is.

7. PLOS authors have the option to publish the peer review history of their article (what does this mean? ). If published, this will include your full peer review and any attached files.

**Do you want your identity to be public for this peer review?** For information about this choice, including consent withdrawal, please see our Privacy Policy .

Reviewer #1: No

---

## [Editor Report · Acceptance letter]

PONE-D-25-14407R1

PLOS ONE

Dear Dr. Cao,

I'm pleased to inform you that your manuscript has been deemed suitable for publication in PLOS ONE. Congratulations! Your manuscript is now being handed over to our production team.

Kind regards,

on behalf of

Professor Dr. Amitava Mukherjee

Academic Editor

PLOS ONE